# The Significance of Preoperative Neutrophil-to-Lymphocyte Ratio (NLR), Platelet-to-Lymphocyte Ratio (PLR), and Systemic Inflammatory Index (SII) in Predicting Severity and Adverse Outcomes in Acute Calculous Cholecystitis

**DOI:** 10.3390/jcm12216946

**Published:** 2023-11-06

**Authors:** Dragos Serban, Paul Lorin Stoica, Ana Maria Dascalu, Dan Georgian Bratu, Bogdan Mihai Cristea, Catalin Alius, Ion Motofei, Corneliu Tudor, Laura Carina Tribus, Crenguta Serboiu, Mihail Silviu Tudosie, Denisa Tanasescu, Geta Vancea, Daniel Ovidiu Costea

**Affiliations:** 1Faculty of Medicine, Carol Davila University of Medicine and Pharmacy Bucharest, 020021 Bucharest, Romania; dragos.serban@umfcd.ro (D.S.); bogdan.cristea@umfcd.ro (B.M.C.); catalin.alius@umfcd.ro (C.A.); ion.motofei@umfcd.ro (I.M.); crenguta.serboiu@umfcd.ro (C.S.); geta.vancea@umfcd.ro (G.V.); 2Fourth General Surgery Department, Emergency University Hospital Bucharest, 050098 Bucharest, Romania; 3Faculty of Medicine, University Lucian Blaga Sibiu, 550169 Sibiu, Romania; dan.bratu@ulbsibiu.ro; 4Department of Surgery, Emergency County Hospital Sibiu, 550245 Sibiu, Romania; 5Department of General Surgery, Emergency Clinic Hospital “Sf. Pantelimon” Bucharest, 021659 Bucharest, Romania; 6Faculty of Dental Medicine, Carol Davila University of Medicine and Pharmacy Bucharest, 020021 Bucharest, Romania; 7Department of Internal Medicine, Ilfov Emergency Clinic Hospital Bucharest, 022104 Bucharest, Romania; 8Department of Nursing and Dentistry, Faculty of General Medicine, University Lucian Blaga Sibiu, 550169 Sibiu, Romania; denisa.tanasescu@ulbsibiu.ro; 9Clinical Hospital of Infectious and Tropical Diseases “Dr. Victor Babes”, 030303 Bucharest, Romania; 10Faculty of Medicine, Ovidius University Constanta, 900470 Constanta, Romania; daniel.costea@365.univ-ovidius.ro; 11General Surgery Department, Emergency County Hospital Constanta, 900591 Constanta, Romania

**Keywords:** acute cholecystitis, systemic inflammatory biomarkers, NLR, PLR, SII, postoperative outcome

## Abstract

The prediction of severity in acute calculous cholecystitis (AC) is important in therapeutic management to ensure an early recovery and prevent adverse postoperative events. We analyzed the value of the neutrophil-to-lymphocyte ratio (NLR), platelet-to-lymphocyte ratio (PLR), and systemic inflammatory index (SII) to predict advanced inflammation, the risk for conversion, and postoperative complications in AC. Advanced AC was considered the cases with empyema, gangrene, perforation of the gallbladder, abscesses, or difficulties in achieving the critical view of safety. A 3-year retrospective was performed on 235 patients admitted in emergency care for AC. The NLR was superior to the PLR and SII in predicting advanced inflammation and risk for conversion. The best predictive value was found to be at an NLR “cut-off” value of >4.19, with a sensitivity of 85.5% and a specificity of 66.9% (AUC = 0.824). The NLR, SII, and TG 13/18 correlate well with postoperative complications of Clavien–Dindo grade IV (*p* < 0.001 for all variables) and sepsis. For predicting early postoperative sepsis, TG 13/18 grading >2 and NLR > 8.54 show the best predicting power (AUC = 0.931; AUC = 0.888, respectively), although not significantly higher than that of the PLR and SII. The NLR is a useful biomarker in assessing the severity of inflammation in AC. The SII and PLR may be useful in the prediction of systemic inflammatory response.

## 1. Introduction

Acute calculous cholecystitis (ACC) is a common cause of abdominal pain in emergencies. In a multicentric study designed by the World Society of Emergency Surgery in 2015, ACC ranked as the second cause of complicated intra-abdominal infections, accounting for 18.5% of the total number of cases [1,2]. Early laparoscopic cholecystectomy (LC) is the gold standard in the current therapeutical approach, with favorable outcomes in most cases. However, recent studies found a 0.1–1% mortality risk and a 6–9% risk of major complications [3], such as main common bile duct lesions, myocardial infarction, and pulmonary complications, and this risk is highly increased in emergency LC performed in cases with severe inflammation. A recent study by Lucocq et al. [4] found that 36.7% of LCs performed in emergency had a non-standard outcome, including conversion, subtotal cholecystectomy, bile leak, and prolonged postoperative stay [4]. Most of these cases were related to the severity of local inflammation, intraoperative findings showing gangrenous cholecystitis, empyema, perforation of the gallbladder, and difficult dissection of the Calot Triangle, all these conditions being generally referred to as advanced acute cholecystitis [5] or severe cholecystitis [6,7] by different authors. Preoperative identification of these cases is important to optimize the therapeutic approach and improve the clinical outcome [5].

Currently, the role of different biomarkers with predictive value in acute cholecystitis is still a subject of research. TG13/18 guidelines propose a grading scale for evaluating local inflammation and its systemic involvement based on clinical evaluation, leukocytes, and CRP as well as the presence or not of the alterations of the vital functions related to the septic process [8,9].

However, a systematic review of Tufo et al. [3] found that while grade III TG 13/18 may be associated with higher mortality when compared with grade I, there is no consensus regarding the preoperatory predictive risk evaluation in patients with acute calculous cholecystitis. 

Several studies found CRP to be a good predictive factor for conversion; however, the cut-off values varied widely from 76 mg/L to 220 mg/L [5,10,11,12,13]. Together with the valuable findings provided by ultrasound and CT exam, specific biomarkers were analyzed for the possible predictive role for the severity of local inflammation, such as the YKL-40 protein level [14], serum level of visfatin [15], procalcitonin [16], human neutrophil lipocalin [17], chitotriosidase, and neopterin [18]. However, their availability in emergencies is limited in many surgical departments.

Recently, the systemic inflammatory biomarkers, neutrophil-to-lymphocyte ratio (NLR) and platelet-to-lymphocyte ratio (PLR), were investigated for their predictive value in many inflammatory and septic conditions [18,19,20,21,22], such as septic shock, diabetic foot ulcer, acute appendicitis, and spontaneous bacterial peritonitis. They are cheap and inexpensive biomarkers, easy to calculate based on the complete blood count (CBC). Several studies found a good correlation between the NLR and PLR and the severity of inflammation in acute calculous cholecystitis as well as the length of postoperative stay. However, there is still conflicting evidence regarding the clinical significance of these biomarkers and their cut-off value that could be used in therapeutic management.

In the present study, we aimed to analyze the value of the NLR, PLR, and SII in predicting severe forms of acute cholecystitis, conversion to open surgery, and adverse postoperative outcomes.

## 2. Materials and Methods

### 2.1. Patient Selection

A 3-year retrospective study was carried out between January 2020 and December 2022 on the patients admitted for acute cholecystitis in the 4th Department of Surgery, Emergency University Hospital Bucharest. Data were collected from electronic patient records and operatory protocols. All patients admitted in emergency care, aged over 18 years, for whom the diagnosis of acute cholecystitis could be confirmed based on intraoperative findings were included in the statistical analysis. Along with local and systemic inflammatory signs, ultrasonography and/or abdominal CT were used to document the presence of calculi, the thickness of the gallbladder walls, common biliary duct (CBD) diameter, and the potential signs of pericholecystitis. For all patients, age, associated comorbidities, time elapsed from the onset of symptoms to presentation, and clinical signs were assessed at admission. Biological tests at admission included a complete blood count with differentials of fibrinogen, bilirubin, hepatic transaminases, INR, urea, and creatinine. Systemic inflammatory biomarkers were calculated based on the counts for neutrophils, platelets, and neutrophils measured from the same sample and expressed as their value in cells/L. SII was calculated using the formula SII = P × N/L, where P, N, and L are the counts of platelets, neutrophils, and lymphocytes, respectively [23].

C-reactive protein (CRP) was not available in an emergency in our hospital but was determined the next day, in cases in which surgical intervention was postponed due to local or general conditions. For this reason, CRP was not included in the statistical analysis.

Patients with associated malignancies as well as hematological and autoimmune diseases were excluded due to their previously documented impact on the blood cells and derivate systemic inflammatory indices.

### 2.2. Study Design

The patients included in the study were classified according to the intraoperative findings into mild and advanced acute cholecystitis, according to the intensity of local inflammation. Advanced forms were considered the cases with empyema, gangrene, perforation of the gallbladder, abscesses, adhesions, or difficulty in dissecting Calot’s triangle, likely to be associated with increased operative difficulty [9,24].

The patients were classified according to TG 13/TG18 Tokyo guidelines for acute cholecystitis as grade I (mild) acute cholecystitis, grade II (moderate), and grade III (severe) if associated with organ dysfunction [9]. Systemic inflammatory biomarkers NLR, PLR, and SII were calculated based on the complete blood cell count at admission. The prediction values of TG 13/TG 18 severity grading, NLR, PLR, and SII were analyzed for advanced AC, postoperative complications, and hospital stay.

### 2.3. Statistical Analysis

Microsoft Excel and Med Calc^®^ Statistical Software (version 22.006 Med Calc Software Ltd., Ostend, Belgium; https://www.medcalc.org; accessed on 10 August 2023) were used for data analysis. Pearson’s Chi-squared test was used to evaluate the association between discrete variables, while ANOVA was used for continuous variables. For the statistically significant results, a post hoc analysis was performed to establish the differences within groups by using the Scheffe test for all pairwise comparisons.

The specificity and sensitivity of NLR, PLR, and SII in predicting the severity of inflammation, and local and systemic complications were analyzed by ROC curves. According to the widely accepted classification scale described by Safari et al. [25], the AUC values were categorized as 90–100 = excellent; 80–90 = good; 70–80 = fair; 60–70 = poor; and 50–60 = fail.

## 3. Results

### 3.1. General Data of the Patients Included in the Study Group

A total of 235 patients with acute cholecystitis were included in the study, with a mean age of 54.6 ± 16.3. Most of the cases were mild (70.6%) and of female patients (71.4%). In the advanced AC group, the mean age was significantly higher (61 ± 15.6 vs. 52 ± 15.9, <0.001), and there were significantly more male patients (*p* = 0.008) when compared to mild cases (Table 1).

Most patients included in the study group presented with two or more comorbidities. The subjects included in the advanced AC group had significantly more comorbidities than those admitted with mild AC (*p* < 0.001). Older age (*p* < 0.001), obesity (*p* = 0.047), diabetes (*p* = 0.001), ischemic cardiac disease (*p* = 0.01), chronic hepatic diseases (*p* = 0.02), and cardiac failure/shock at admission (*p* = 0.01) were correlated with advanced AC in the study group. According to the ASA risk scale, most patients were graded as grade II or III in both groups. However, there was an upward trend of distribution towards higher grades in the advanced AC group, confirmed by the linear-by-linear association test (*p* = 0.0003). A similar upward trend was observed for the TG 13/18 severity scale, with more grade II and III cases in the advanced AC group (<0.0001).

Statistical analysis showed significantly higher values for leukocytes (*p* < 0.0001), neutrophils (*p* = 0.001), the NLR (*p* < 0.001), the PLR (*p* < 0.001), the SII (*p* < 0.001), fibrinogen (*p* < 0.001), and bilirubin (*p* < 0.001) with no significant difference for platelets, INR, transaminases, and creatinine levels. 

### 3.2. Comparative Analysis of NLR, PLR, and SII Values with TG 13/18 Grading in the Study Group

Furthermore, we investigated how the TG13/18 severity grading scale for AC correlates with the values of NLR, PLR, and SII by using the Chi-squared test and Scheffe test for pairwise comparison. The statistical analysis found a significant positive correlation in all cases, with the mean values of the investigated systemic inflammatory biomarkers rising from the grade I to grade III groups. However, there are differences in the Scheffe test results, which may suggest that each of the biomarkers characterizes specific changes in the inflammation process (Table 2).

While the NLR is an early inflammatory biomarker, which significantly raises between mild and moderate forms, the PLR seems to be significantly elevated in advanced stages, when local inflammation of the gallbladder and surrounding tissues reaches systemic involvement. SII values, combining in their formula both the number of neutrophils and platelets, discriminate best among the three stages defined by the TG 13/18 scale.

### 3.3. Prediction Value of NLR, PLR, SII, and TG 13/18 Grading Scale for Advanced Acute Cholecystitis

The sensitivity and specificity of the NLR, PLR, SII, total leukocytes, and TG 13/18 grading scale for predicting advanced forms of AC were analyzed by the ROC curves (Figure 1).

Only the NLR showed a good predictive value (AUC = 0.824). The pairwise comparison of ROC curves for predicting advanced AC found that the predictive value of the NLR was significantly superior to that of the SII (*p* = 0.0065), PLR (*p* < 0.0001), total leukocytes (*p* = 0.0103), and TG 13/18 grading (*p* = 0.0006). The best predictive value was found to be at a cut-off value of >4.19, with a sensitivity of 85.5% and a specificity of 66.9% (Table 3).

### 3.4. Surgical Approach and Postoperative Outcomes

Laparoscopic cholecystectomy was the most common procedure in both groups. However, the number of cases that required conversion, open surgical procedures, Kehr drainage, or perioperative ERCP was significantly higher in the advanced AC group (Table 4).

The reason for conversion to open surgery in the mild AC group was the unclear anatomy of the Calot triangle due to extensive fibrosis. In the advanced AC group, most cases were converted due to a friable hemorrhagic gangrenous gallbladder wall (five cases) and the impossibility of achieving the critical view of safety (CVS) due to inflammation and adherences (six cases). Other causes of conversion included biliary fistula (one case), Mirizzi Syndrome (one case), biliary peritonitis due to a perforated gallbladder abscess (one case), and a pericholecystic abscess (one case).

Open surgery as the first choice was mainly dictated by the general status and associated comorbidities in patients graded as ASA IV or V (seven cases, including the three cases in the mild AC group), for whom the laparoscopic approach was not considered safe to be performed by the intensive care team. In three cases, the decision was made based on the clinical and imagistic data: pseudo-tumoral pericholecystic mass (one case) and gallbladder abscess (two cases). There was one case treated by cholecystostoma, an 85-year-old patient with piocholecystitis and septic shock at admission, who died 3 days after surgery in the intensive care unit due to sepsis and acute limb ischemia.

In the study group, there were 16 patients who were COVID-19-positive at the moment of admission. Out of these, 14 were treated safely by laparoscopic cholecystectomy, after all the required safety measures were taken to prevent the contamination of the operatory team. In the remaining cases, open surgery was performed due to associated septic shock (one case) and COVID-19 severe pneumonia (one case).

Furthermore, we analyzed the postoperative complications encountered in the study group, registered after Clavien–Dindo classification (Table 5). 

Statistically significant differences observed between the mild and advanced AC groups for surgical site infections (*p* = 0.043) and nosocomial infections (*p* = 0.007) could be correlated with higher numbers of open surgeries and conversions, as well as with increased hospital stays in the advanced AC group. General complications requiring intensive care were more frequent in the advanced AC group (*p* = 0.002), including sepsis (*p* = 0.004) and postoperative malign hypertension (*p* = 0.043).

### 3.5. Correlations between Inflammatory Parameters and Types of Surgery in the Study Group

NLR and TG13/18 grading correlated well with the type of surgery performed (*p* = 0.001; and *p* < 0.0001, respectively), while the PLR and SII mean values were higher in the conversion and open surgery groups but not statistically significant (Table 6). 

The predictive value for conversion for the NLR, PLR, SII, TG 13/18, and total leukocytes was analyzed by ROC curves (Figure 2).

Out of the studied parameters, only the NLR showed a good predictive value for conversion, with a cut-off value of 4.24 (AUC = 0.802, *p* < 0.001), significantly higher compared to that of leukocytes (AUC = 0.755), SII (AUC = 0.734), and TG13/18 (AUC = 0.690) (Table 7).

### 3.6. Correlations between Inflammatory Parameters and Postoperative Outcomes in the Study Group

ANOVA showed a good correlation between the NLR, PLR, SII, and TG 13/18 grading scale and the postoperative hospital stay (*p* < 0.001; *p* < 0.001; *p* < 0.001; and *p* = 0.008, respectively) and total hospital stay (*p* = 0.002; *p* < 0.001; *p* < 0.001; and *p* = 0.001, respectively).

In the present study, the NLR, PLR, SII, and TG 13/18 grading scales had a poor prognostic value for predicting local postoperative complications, almost equal to a coin toss (Table 8, Figure 3), and did not correlate well with the postoperative complications related to surgery, Clavien–Dindo grades II and III (*p* = 0.83; *p*= 0.843; and *p* = 0.898, respectively). 

However, the NLR, SII, and TG 13/18 correlated well with postoperative complications of Clavien–Dindo grade IV (*p* < 0.001 for all variables), while the values were not statistically significant for the PLR (*p* = 0.113). However, their predictive power evaluated by ROC curves varied from poor (PLR and SII) to fair (TG 13/18 grading and NLR), as shown in Table 9, Figure 4. 

For predicting early postoperative sepsis, a TG 13/18 grading > 2 and NLR > 8.54 showed the best predicting power (AUC = 0.931; AUC = 0.888, respectively), though not significantly higher than that of the PLR and SII (Table 10, Figure 5).

## 4. Discussion

Predicting the severity of acute cholecystitis is important to achieve the best therapeutic outcomes and prevent adverse postoperative events [2,26,27,28,29]. Local inflammation and surgical trauma induce metabolic and systemic inflammatory responses, which may lead to systemic complications [30]. Understanding and addressing inflammation and the possible systemic imbalances that it may cause is important to prevent adverse outcomes and unnecessarily prolonged hospital stays in cases with AC. 

Commonly used for diagnosis, CT and ultrasound examination may not accurately predict advanced AC [30,31]. In a study on 1115 patients who underwent surgery for acute calculous cholecystitis, Goiayev et al. [31] found that even in cases with a gallbladder wall of ≤ 4.85 mm, if the NLR > 5.65 and the total leukocytes exceed 8100/mm^3^, there is a 92% probability of complicated AC, including gangrenous, perforated, emphysematous, or necrotizing AC. NLR is a cheap, easy-to-calculate inflammatory biomarker that combines the relative ratio of neutrophils—the first line of cellular defense in acute inflammation—and the lymphocytes, with an immunomodulatory role [30].

Although several studies have found a significant correlation between the NLR and the severity of inflammation in AC [30,32,33,34], there is limited evidence regarding the specific cut-off value, with possible clinical use.

In the present study, we comparatively examined NLR, PLR, SII, total leukocytes, and TG13/18 grading scale for predicting severe inflammation in acute cholecystitis, risk for conversion, and adverse outcomes. We found that the NLR performed best for predicting advanced AC, with an AUC of 0.824 at a cut-off value >4.19. The NLR also has a good predictive value for conversion (AUC = 0.804, cut-off value of 4.24), with high sensitivity (93.7%) but low sensitivity (55.7%).

A previous study by Micic et al. [24] on 136 patients who underwent LC for acute cholecystitis found a similar cut-off value of 4.18 for predicting advanced AC with a 78.3% sensitivity and 74.3% specificity [24], while another recent study found a cut-off value of 4.17 for moderate to severe AC [35], with a predictive value similar to that of CRP. 

A higher “cut-off” value of 5.5 for the NLR was found by Turhan et al. [36] with a good predictive value, 80.8% sensitivity, and 80.1% specificity. This may be explained, however, by the selection criteria the authors defined for the complicated AC group in their study, which included very advanced changes of the gallbladder wall, such as perforation, gangrenous cholecystitis, and emphysematous cholecystitis. The definition of “difficult cholecystectomy” is still a challenging subject, with no international consensus being reached. In the present study, we followed the recommendations of Manuel Velasques et al. [37], so we also included severe local inflammation which led to the impossibility of achieving the critical view of safety. On the other hand, Turhan et al. [37] also found that the PLR correlated with inflammation, but with a lower predictive value when compared to that of the NLR for complicated AC (AUC = 0.704 vs. 0.873, respectively), which is consistent with our findings. Diez Ares et al., in a study on 130 patients operated on for AC, found that an NLR value of >5 and a CRP value of >100 mg/dL were independent risk factors for gangrenous cholecystitis, with good predictive value estimated by ROC curves (AUC = 0.75 vs. AUC = 0.80, respectively), and should be taken into account in the therapeutic decision, considering that early laparoscopic cholecystectomy provides the best outcomes in gangrenous AC [38].

A different approach was used by Unal et al. [34], who analyzed the correlation between the NLR and the TG 13/18 grading scale. He found that an NLR cut-off value of 5.2 may discriminate well between TG13/18 grade 1 vs. grades 2 and 3 with a sensitivity of 76.76% and specificity of 76.17% (AUC of 0.817), while a NRL > 8.5 is a good predictor for TG 13/18 grade 3, which associates with systemic imbalance due to inflammation [34]. In our study, we also found a cut-off value of >8.54 to be a good predictor for early postoperative sepsis.

Kartal and Kalayci [38] found no correlation between the NLR and postoperative overall morbidity in the elderly with AC [38]. In the present study, we found no correlation between the systemic inflammatory biomarkers and Clavien–Dindo complications grades II and III. This finding may support the current recommendation that early cholecystectomy may be performed safely in all cases of acute cholecystitis, even those with severe inflammation. However, inflammatory biomarkers were well-correlated with the grade IV Clavien–Dindo complications, requiring intensive care. In the present study, we found that an NLR value of >8.54 has 87.5% sensitivity and 81% specificity for early postoperative sepsis. Although postoperative complications are more frequent in the severe cholecystitis group, there is no correlation between postoperative surgical complications and the values of the NLR, SII, PLR, or TG 13/18 grading scale. This supports, on the one hand, the idea that choosing the appropriate technique for each case allows for a successful solution regardless of the severity of the local inflammation [1,39,40,41]. On the other hand, repeated inflammatory relapses that lead to local fibrous rearrangements, but also the involvement of the human factor (perception errors of the struts during dissection), can generate vascular-biliary lesions [42,43].

In our study, we found that the PLR is an important biomarker in predicting sepsis in patients with AC admitted in emergency care. Part of the complicated underlying pathophysiology of sepsis syndrome is clot formation and bleeding diathesis associated with platelet disfunction, endothelial activation, and disseminated intravascular coagulation [44]. Prompt identification of these patients is essential for improving the survival rate in these patients [44,45,46].

Mitigation of perioperative inflammation and pain is important for enhanced recovery after surgery (ERAS) and preventing postoperative complications [47,48,49]. Postoperative analgesia is a very important part of perioperative management in patients with AC. The pain pattern after LC seems to be different from that after other laparoscopic surgeries [48], and good pain management should be based on an individualized approach. The intensity of preoperative inflammation may sensibilize the peritoneal nociceptors, so a multimodal analgesia could be the best option to control the pain with minimal adverse effects [48]. Several studies found that choosing non-opioid combinations, such as paracetamol and parecoxib 40 mg IV or lornoxicam quick-release 8 mg PO every 12 h, results in the same anti-algic effect as opioids, but limits the risk of pulmonary complications and allows a quick recovery [50,51].

Our study has some limitations: it is a monocentric retrospective study on a limited number of patients. The analyzed values are those from admission, not those from the operative moment. Also, the dynamics of biomarkers regarding the development of postoperative complications were not analyzed. We also could not differentiate between the cases that needed conversion due to sclerosis after multiple previous episodes of mild AC and the impossibility of achieving the critical view of safety due to active inflammation. This might be an explanation for the lower cut-off value found for the NLR when compared to other studies that focus on the gangrenous gallbladder only. However, our study brings valuable information regarding the correlations between the NLR, PLR, and SII and the severity of AC, risk for conversion, and postoperative morbidity.

## 5. Conclusions

The NLR, PLR, and SII are useful in the preoperative assessment of the AC. The NLR is an early biomarker of inflammation, with higher predictive value when compared to that of PLR, SII, and total leukocytes, and is more versatile than the TG 13/18 scale, being a continuous variable. An NLR value of >4.19 is suggestive of advanced inflammation, while a value of >8.54 is a good predictor for early postoperative sepsis. The PLR and SII correlate significantly with the severity of the inflammation and may be useful in the prediction of the systemic inflammatory response, but they have fair predictive value for advanced AC and risk for conversion in LC.

## Figures and Tables

**Figure 1 jcm-12-06946-f001:**
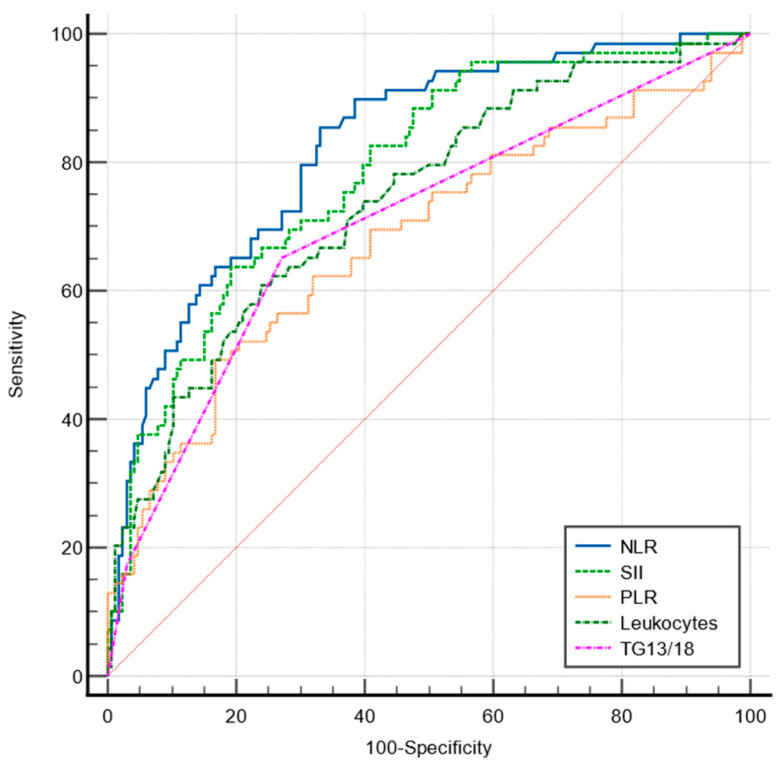
Comparative ROC curves for NLR, PLR, SII, TG13/18, and total leukocytes in predicting advanced AC.

**Figure 2 jcm-12-06946-f002:**
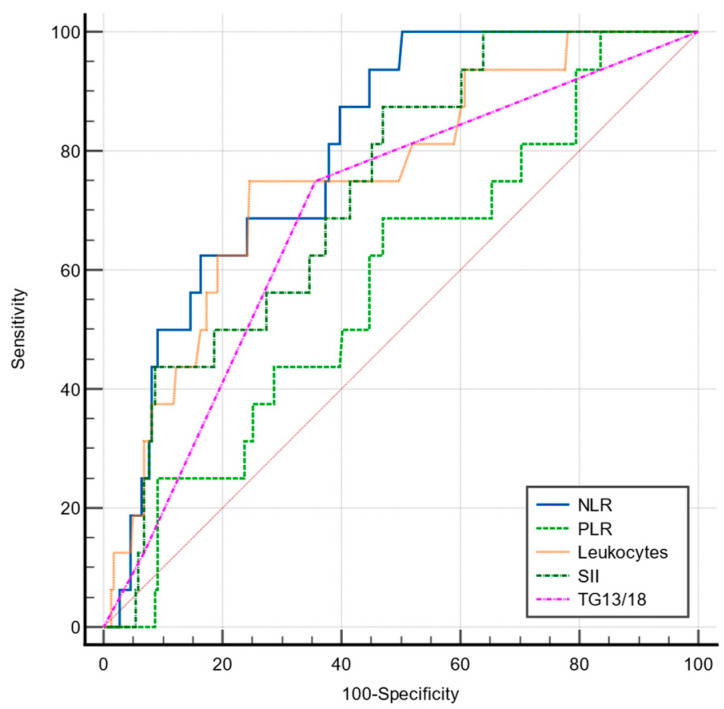
Comparative ROC curves for NLR, PLR, SII, TG13/18, and total leukocytes in predicting conversion to open cholecystectomy.

**Figure 3 jcm-12-06946-f003:**
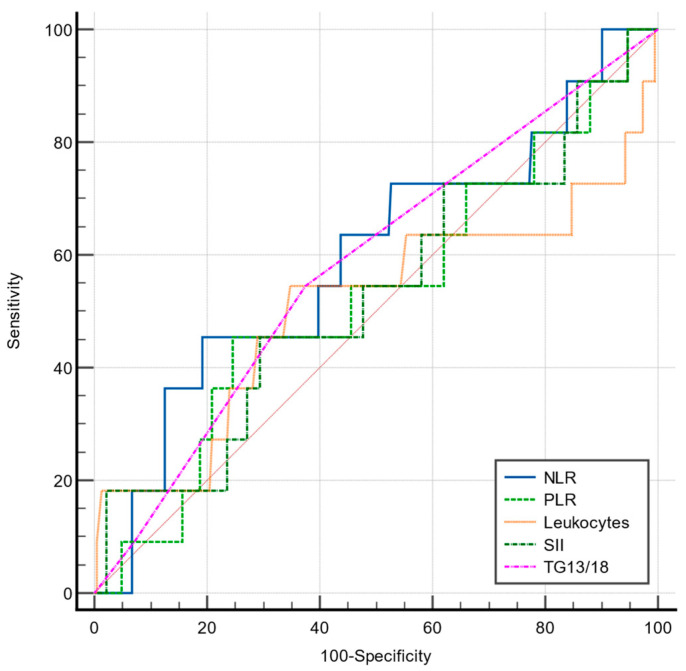
Comparative ROC curves for NLR, PLR, SII, TG13/18, and total leukocytes in predicting postoperative local complications.

**Figure 4 jcm-12-06946-f004:**
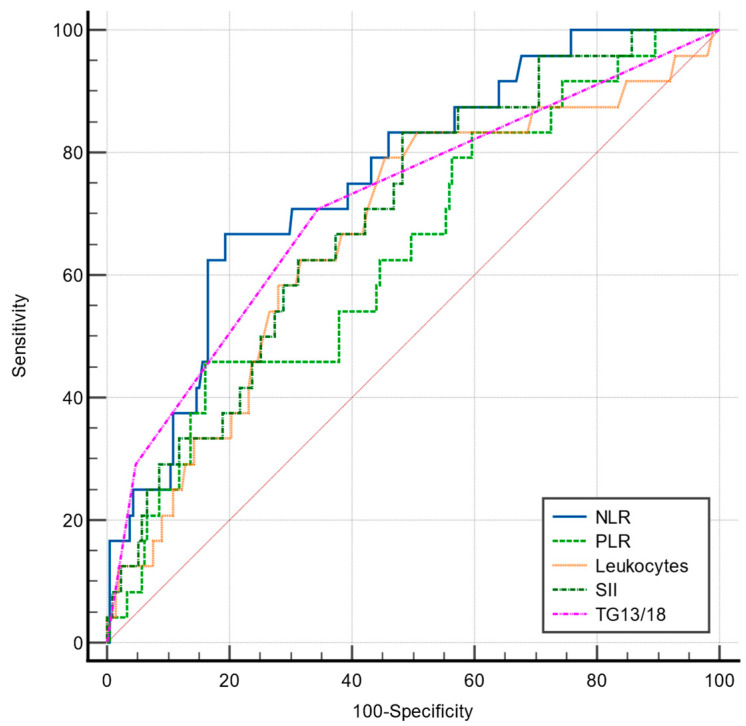
Comparative ROC curves for NLR, PLR, SII, TG13/18, and total leukocytes in predicting general complications requiring intensive care support (Clavien–Dindo grade IV).

**Figure 5 jcm-12-06946-f005:**
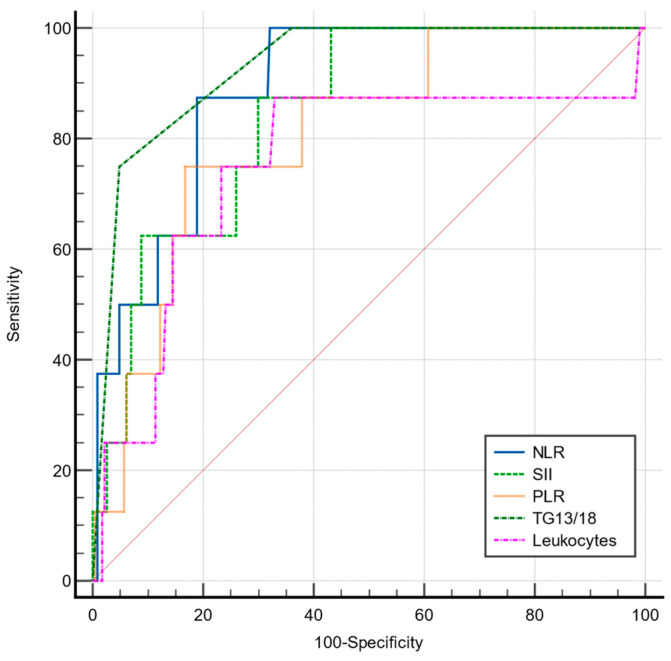
Comparison of ROC curve for NLR, SII, PLR, TG13/18, and total leukocytes in predicting sepsis in the early postoperative period.

**Table 1 jcm-12-06946-t001:** General data of the patients included in the study group.

**Parameter**	**Total**	**Mild AC**	**Advanced AC**	***p* Value**
No. of patients	235	166 (70.6%)	69 (29.4%)	
Females	168 (71.4%)	127 (76.5%)	41 (59.4%)	0.008 ^1^
Age	54.6 ± 16.3	52 ± 15.9	61 ± 15.6	<0.001 ^1^
Comorbidities (No.): Obesity Arterial hypertension Cardiac ischemic disease Chronic hepatic diseases Chronic respiratory diseases Chronic renal diseases Cardiac failure/shock Diabetes Others	2 ± 1.4	1.8 ± 1.3	2.6 ± 1.6	<0.001 ^2^
	116 (49.4%)	75 (45.1%)	41 (59.4%)	0.047 ^2^
	107 (45.5%)	57 (53.3%)	50 (46.7%)	0.093 ^2^
	28 (11.9%)	14 (8.4%)	14 (20.2%)	0.01 ^2^
	67 (28.5%)	40 (24%)	27 (39.1%)	0.02 ^2^
	31 (13.2%)	18 (10.8%)	13 (18.8%)	0.099 ^2^
	39 (16.6%)	25 (15%)	14 (20.2%)	0.327 ^2^
	11 (4.7%)	4 (2.4%)	7 (10.1%)	0.01 ^2^
	32 (13.6%)	15 (9%)	17 (24.6%)	0.001 ^2^
	84 (35.7%)	62 (37.3%)	22 (31.8%)	0.426 ^2^
ASA PS risk scale I II III IV V				0.008 ^2^ (0.0003 ^3^ for trend)
	16 (6.8%)	14 (8.4%)	2 (2.8%)	
	124 (52.8%)	96 (57.8%)	28 (40.5%)	
	78 (33.2%)	48 (28.9%)	30 (43.4%)	
	16 (6.8%)	8 (4.8%)	8 (11.5%)	
	1 (0.4%)	0	1 (1.4%)	
TG 13/18 severity grading I II III				<0.0001 ^2^ (<0.0001 ^3^ for trend)
	145 (61.7%)	121 (72.9%)	24 (37.4%)	
	73 (31.1%)	40 (24%)	33 (47.8%)	
	17 (7.2%)	5 (3%)	12 (17.4%)	
Angiocholitis/CBD stones	18 (7.6%)	7 (4.2%)	11 (15.9%)	0.013 ^2^
Leukocytes (/μL)	10,441 ± 4895.3	9187.6 ± 3787.4	13,456.2 ± 5882	<0.0001 ^1^
Neutrophils (/μL)	7796 ± 4867.5	6413.4 ± 3728.6	11,124.9 ± 5646.6	0.001 ^1^
Platelets (/μL)	239,767.1 ± 82,016.7	245,341.4 ± 77,532.9	226,356.5 ± 91,122	0.053 ^1^
Fibrinogen (mg/dL)	450.1 ± 186.2	389.1 ± 119.1	596.8 ± 232.3	<0.001 ^1^
INR	1.3 ± 1.1	1.2 ± 0.9	1.3 ± 1.2	0.327 ^1^
Bilirubin	1.3 ± 2.0	0.95 ± 1.1	2.3 ± 3.1	<0.001 ^1^
AST	68.9 ± 116	63.2 ± 116.2	125 ± 297.1	0.063 ^1^
ALT	107.4 ± 165.9	85.8 ± 136.9	120 ± 179	0.056 ^1^
Creatinine	1.3 ± 0.5	1.2 ± 0.3	1.5 ± 1.4	0.341 ^1^
NLR	7.29 ± 12.2	4.3 ± 5.2	14.3 ± 19.4	<0.001 ^1^
PLR	181.2 ± 229.4	143.8 ± 68.7	273.5 ± 397	<0.001 ^1^
SII	1701.6 ± 3416.4	1009.5 ± 993.8	3366.8 ± 5812.4	<0.001 ^1^

Footnote: ^1^ ANOVA; ^2^ Chi-squared test; ^3^ test of linear-by-linear association; ASA PS: American Society of Anesthesiologists Physical Status Classification; TG13/18: Tokyo Guidelines classification risk; AST: aspartate aminotransferase; ALT: Alanyl aminotransferase; NLR: neutrophil-to-lymphocyte ratio; PLR: platelet-to-lymphocyte ratio; SII: systemic inflammatory index.

**Table 2 jcm-12-06946-t002:** Correlations between NLR, PLR, and SII with TG 13/18 grading in the study group.

	TG13/18 Grade I (1)	TG13/18 Grade II (2)	TG13/18 Grade III (3)	*p* Value (Chi-Squared Test)	Scheffe Test for Pairwise Comparison
NLR	3.6 ± 3	11.7 ± 14.6	18.8 ± 28	<0.001	(1) differs from (2) and (3)
PLR	147.8 ± 80.3	191.2 ± 123.6	432.1 ± 752	<0.001	(1) and (2) differ from (3)
SII	879.9 ± 726.5	2393.6 ± 2477	5738.6 ± 10,617	<0.001	Each group differs significantly from the others.

**Table 3 jcm-12-06946-t003:** Sensitivity and specificity at the “cut-off” value predicting advanced forms.

	Sensitivity	Specificity	Cut-Off Value	AUC	*p*
NLR	85.5	66.9	>4.19	0.824	<0.001
PLR	49.3	83.1	>189.3	0.679	<0.001
SII	63.8	80.7	>1442.4	0.787	<0.001
TG13/18	65.22	72.8	>1(mild)	0.704	<0.001
Leukocytes	60.87	75.9	>11,300	0.741	<0.001

**Table 4 jcm-12-06946-t004:** Surgical treatment and outcomes.

	Total (n = 235)	Mild AC (n = 166)	Advanced AC (n = 69)	*p* Value
Types of surgery: LC LC-conversion CC Cholecystostomy				<0.0001 ^1^ (<0.0001 ^2^ for trend)
	208 (88.6%)	162 (97.6%)	46 (66.6%)	
	16 (6.8%)	1 (0.6%)	15 (21.7%)	
	10 (4.2%)	3 (1.8%)	7 (10.2%)	
	1 (0.4%)	0	1 (1.5%)	
Kehr drainage	5 (2.1%)	1 (0.6%)	4 (5.8%)	0.012 ^1^
ERCP + calculi removal (pre or postop)	17 (7.2%)	7 (4.2%)	10 (14.5%)	0.005 ^1^
Postoperative hospital stay (days)	3.6 ± 3.4	2.9 ± 2.8	5.1 ± 4	<0.001 ^3^
Length of stay (days)	7.1 ± 4.5	6.1 ± 3.9	9.3 ± 5.2	<0.001 ^3^

Footnote: LC = laparoscopic cholecystectomy; CC = classic (open) cholecystectomy; ERCP = endoscopic retrograde cholecysto-pancreatography; ^1^—Chi-squared test; ^2^—Scheffe test for pairwise comparison; ^3^—ANOVA.

**Table 5 jcm-12-06946-t005:** Postoperative complications according to Clavien–Dindo Classification.

	Total (n = 235)	Mild AC (n = 166)	Advanced AC (n = 69)	*p*-Value *
I (surgical site infections)	4 (1.7%)	1 (0.6%)	3 (4.3%)	0.043
II (requiring pharmacological treatment) surgical-related complications, treated conservatory Nosocomial infections				

	11 (4.6%)	5 (3%)	6 (8.6%)	0.064


	15 (6.4%)	6 (3.6%)	9 (13%)	0.007
III (surgical-related complications requiring endoscopic/surgical/Rx approach)	2 (0.8%)	1 (0.6%)	1 (1.4%)	1.00
IV (general complications requiring intensive care) Malign hypertension Hemodynamic instability Sepsis Pulmonary edema/pleurisy	16 (6.8%)	4 (2.4%)	12 (17.3%)	0.002

	4 (1.7%)	1 (0.6%)	3 (4.3%)	0.043
	1 (0.4%)	0	1 (1.4%)	0.12
	8 (3.4%)	2 (1.2%)	6 (8.6%)	0.004
	3 (1.3%)	1 (0.6%)	2 (2.8%)	0.15
V (deceased)	5 (2.1%)	2 (1.2%)	3 (4.3%)	0.129

Footnote: * *p*-value was calculated by Chi-squared test.

**Table 6 jcm-12-06946-t006:** Correlations between the types of surgery and NLR, PLR, SII, and TG13/18 in the study group.

	LC (1)	LC-Conversion (2)	CC/Cholecystostomy (3)	*p*-Value	Scheffe Test for Pairwise Comparison
NLR	6.2 ± 11.8	12.1 ± 7	20.2 ± 17.3	0.001 ^1^	(1) differs from (3)
PLR	178.5 ± 241.9	182 ± 82.7	248.1 ± 73.4	0.823 ^1^	NS
SII	1583.8 ± 3572.4	2323.2 ± 1474.2	3014.5 ± 1779.5	0.49 ^1^	NS
TG 13/18 grading				<0.0001 ^2^	Each group differs significantly from the others
I	139 (66.9%)	4 (25%)	2 (18.2%)		
II	58 (27.9%)	10 (62.5%)	5 (45.4%)		
III	11 (5.2%)	2 (12.5%)	4 (36.4%)		

^1^ ANOVA; ^2^ Chi-squared test; NS: not significant.

**Table 7 jcm-12-06946-t007:** Prediction value of NLR, PLR, SII, and TG 13/18 for conversion to open surgery.

	PDR Sensitivity	PDR Specificity	Cut-Off Value	AUC	*p*
NLR	93.7	55.2	>4.24	0.802	<0.001
PLR	68.7	53	>141.8	0.582	0.246
SII	87.5	53	>949.6	0.734	<0.001
TG13/18	75	64.3	>1	0.690	0.001
Leukocytes	75	75.3	>12,200	0.755	<0.001

**Table 8 jcm-12-06946-t008:** Prediction value of NLR, PLR, SII, and TG 13/18 for surgical-related postoperative complications.

	PDR Sensitivity	PDR Specificity	Cut-Off Value	AUC	*p*
NLR	45.45	80.8	>8.88	0.595	0.33
PLR	45.45	75.45	>194.6	0.528	0.776
SII	45.45	70.54	>1525.9	0.530	0.3
TG13/18	54.5	62.5	>1	0.583	0.31
Leukocytes	72.73	5	<17,800	0.510	0.935

**Table 9 jcm-12-06946-t009:** Prediction value of NLR, PLR, SII, and TG 13/18 for general postoperative complications requiring intensive care (Clavien–Dindo IV).

	Sensitivity	Specificity	Cut-Off Value	AUC	*p*
NLR	66.7	80.6	>7.67	0.758	<0.001
PLR	45.8	83.9	>221.3	0.640	0.02
SII	83.3	51.7	>858.3	0.697	0.001
TG13/18	70.83	65.4	>1	0.715	<0.001
Leukocytes	79.2	54.4	>9100	0.668	0.006

**Table 10 jcm-12-06946-t010:** Prediction value of NLR, PLR, SII, and TG 13/18 for postoperative sepsis.

	PDR Sensitivity	PDR Specificity	Cut-Off Value	AUC	*p*
NLR	87.5	81	>8.54	0.888	<0.001
PLR	75	83.2	>222.46	0.807	<0.001
SII	87.5	70.04	>1447.68	0.845	<0.001
TG13/18	75	92.1	>2	0.931	<0.0001
Leukocytes	87.5	66.9	>11,300	0.753	0.025

## Data Availability

The data presented in this study are available on request from the corresponding author. The data are not publicly available due to privacy.

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
