# Peer review of "The Significance of Preoperative Neutrophil-to-Lymphocyte Ratio (NLR), Platelet-to-Lymphocyte Ratio (PLR), and Systemic Inflammatory Index (SII) in Predicting Severity and Adverse Outcomes in Acute Calculous Cholecystitis"

_jcm, 2023, doi:10.3390/jcm12216946_

Round 1

Reviewer 1 Report

Comments and Suggestions for Authors

The article is well written. Following are my observations:

1.     There are some spelling mistakes like Clavien-Dindo is written as Clavier-Dindo.

2.     There are some language errors too loke in line 164 on page 5.

3.     References 8 and 9 are not sited as per requirements of ICJME.

Comments on the Quality of English Language

1.     There are some spelling mistakes like Clavien-Dindo is written as Clavier-Dindo.

2.     There are some language errors too loke in line 164 on page 5

Author Response

Dear Reviewer,

Thank you very much for your useful comments and appreciation of our work. We have carefully revised the manuscript according to your comments and suggestions. We have corrected the English errors in the text. We have also corrected the references 8 and 9.

We hope in this revised version you will find it suitable to be published.

Reviewer 2 Report

Comments and Suggestions for Authors

This manuscript introduces and comprehensively discusses the value of the neutrophils-to-lymphocyte ratio (NLR), platelets-to-lymphocyte ratio (PLR), and systemic inflammatory index (SII) to predict advanced inflammation, the risk for conversion, and postoperative complications in acute calculous cholecystitis.

The topic is original and relevant to the field. There is limited information on this topic in the literature.

There are no further improvements regarding the methodology.

The conclusions are consistent with the evidence and arguments presented as well as summarize the main point of this article. 

References are up-to-date and appropriate

 Figures and tables are well formatted and make the study easy to follow

Minor revision

I would suggest a brief discussion on postoperative analgesia after laparoscopic cholecystectomy. Consider citing: 

https://pubmed.ncbi.nlm.nih.gov/33155461/

I would suggest a brief discussion on the neutrophil to lymphocyte ratio (NLR) and platelet to lymphocyte percentage ratio (PL%R) in patients with sepsis who were initially treated in the Emergency Department and investigate their predictive ability regarding in-hospital mortality.

Consider citing:

https://pubmed.ncbi.nlm.nih.gov/35801063/

Author Response

Dear Reviewer,

Thank you very much for your useful comments and appreciation of our work. We have revised the manuscript according to your recommendations:

We have added a paragraph regarding postoperative analgesia after LC.

We have also added a paragraph regarding NLR and PLR in patients admitted in emergency with sepsis, as suggested, and updated the references accordingly.

We hope in this revised version, you will find our paper suitable to be published.

Reviewer 3 Report

Comments and Suggestions for Authors

I reviewed the article and want to make the following comments.

Acute cholecystitis is the most common complication of gallstones.

We know different markers of inflammatory response, which is important to identify especially in patients with associated comorbid diseases such as this case. Unfortunately, these are elevated mediately and are not useful when several days have passed since the onset of the disease.

There are various works that show the usefulness of using the leukolithiasis count and the neutrophil index as a marker of the severity of the inflammatory condition, which combined with the platelet count and alterations in coagulation can be useful in these cases.

The Tokyo guidelines are the basis for the management of acute cholecystitis.

I consider that the work corroborates what has been published in various published studies.

The statistical analysis is excellent and the results are adequately analyzed statistically.

Author Response

Dear Reviewer,

Thank you very much for your time spent in reviewing our work and your useful comments! We totally agree with your opinion.